# Compliance with Standard Precaution and Its Relationship with Views on Infection Control and Prevention Policy among Chinese University Students during the COVID-19 Pandemic

**DOI:** 10.3390/ijerph19095327

**Published:** 2022-04-27

**Authors:** Winnie Lai Sheung Cheng, Enid Wai Yung Kwong, Regina Lai Tong Lee, Anson Chui Yan Tang, Lokki Lok Ki Wong

**Affiliations:** 1School of Nursing, Tung Wah College, Hong Kong, China; winniecheng@twc.edu.hk (W.L.S.C.); ansontang@twc.edu.hk (A.C.Y.T.); 2School of Nursing, Putian University, Putian 351100, China; enid.kwong@yahoo.com; 3The Nethersole School of Nursing, The Chinese University of Hong Kong, Hong Kong, China; reginalee@cuhk.edu.hk

**Keywords:** COVID-19, emotion-focused coping, infection control practices, perceived stress, relationship-focused coping

## Abstract

Background: COVID-19 has placed tremendous pressure on the global public health system and has changed daily life. Aim: To examine the relationships between the perceived threat, perceived stress, coping responses and infection control practices towards the COVID-19 pandemic among university students in China. Methods: Using a cross-sectional survey, 4392 students were recruited from six universities in two regions of China. Methods: Data were collected via an online platform using self-reported questionnaires. Hierarchical multiple regression analyses were performed to predict the variables on COVID-19 infection control practices. Results: Pearson correlation coefficients showed a significant negative relationship between perceived stress and COVID-19 infection control practices. A significant positive relationship was observed between wishful thinking and empathetic responding, and infection control practices. Hierarchical multiple regression analyses revealed that gender, geographical location, perceived stress and emotion-focused and relationship-focused coping responses were predictors of COVID-19 infection control practices. Conclusions: The findings suggest that university students displayed moderate levels of stress, using wishful thinking and empathetic responses as coping strategies. Counselling services should therefore emphasise reassurance and empathy. Male university students tended to be less compliant with social distancing. Both counselling and public health measures should recognise the importance of gender differences. Nurses should integrate these findings into future health programme planning and interventions.

## 1. Introduction

COVID-19, a novel coronavirus disease first reported in China in December 2019, is placing tremendous pressure on the global public health system [1] and has changed daily life. To combat viral transmission, governments have imposed stringent emergency measures, such as wearing masks, practicing hand hygiene, and social distancing at local and national levels to limit the spread of the virus [2,3,4].

To avoid further disruption in academic study, universities around the world have now started to resume classes on campus. With many college students gathering on campuses, the risk of transmitting the infection through asymptomatic persons infected with the coronavirus cannot be over-emphasised. Resuming classes places greater pressure on the need for COVID-19 prevention, especially on those who do not comply with infection control practices. Understanding the contributing factors to preventive measures against the spread of disease among university students is important.

## 2. Overview of the Theoretical Constructs

### 2.1. Threat and Behavioural Responses

The pandemic has generated tremendous uncertainty and stress, directly affecting mental health [1]. According to the protection motivation theory [5], individuals initiate cognitive appraisal of a threat and coping responses if they encounter danger and implement actions to mitigate the threat [5,6]. A review of the existing literature reveals that individuals are more likely to perform preventive health behaviours if they perceive a health risk [7].

### 2.2. Stress

Stress is a response to external pressure. Stress can be experienced at any time that an individual perceives a threat to their well-being or when they lack the resources to overcome a situation [8]. According to transactional theory, individuals often seek coping strategies to relieve the negative effects of stress according to their appraisal of their situation [9]. Studies have found that individuals who report higher anxiety are more likely to change their behaviour when encountering health threats [9,10].

### 2.3. Ways of Coping

Coping is a conscious effort by individuals to regulate their emotions, cognition, and behaviour in response to stressful situations; it is a dynamic process, and the strategies adopted are dependent on the situation [8]. People are more likely to use problem-focused coping strategies when they perceive that a stressful situation is within their control and can be changed. Conversely, individuals often use emotion-focused types of coping, such as wishful thinking—an avoidance type of coping—to lessen their emotional distress when they believe a stressful situation cannot be changed [11]. They will use relationship-focused coping to maintain social relationships during a stressful situation [12].

Given that COVID-19 is beyond any individual’s control, avoidance type of coping is likely to be used. Studies have identified several factors, such as feelings of distress, attitude and intention to comply with health regulations that could increase preventive behaviour [13] and the effects of coping strategies on reducing the stress triggered by the pandemic [14].

## 3. Objectives of the Study

Whether avoidance type of coping is effective in initiating preventive practice is scarcely studied. This study postulates that when people perceive stressful situations as beyond their control, they activate coping processes, including wishful thinking and empathy, to mitigate the health threat and stressor. The objectives of this study were: (a) to investigate the relationships between the perceived threat, perceived stress, ways of coping, and infection control practices, and (b) to identify the factors that could predict COVID-19 infection control practices.

## 4. Methods

### 4.1. Study Design and Settings

A cross-sectional study using convenience sampling was conducted. Three higher education institutions in Hong Kong (Special Administrative Region, China) and three higher education institutions in Putian (prefecture-level city in eastern Fujian province, China) were sampled. An invitation letter was sent to the students to fill in the anonymous online questionnaire via mass school emails in April 2020.

### 4.2. Participants

Students aged 18 or older and currently enrolled in a full-time programme offered in the participating education institutions for this study were recruited. Data were collected from 4392 college students (81.8% female; average age, 20.5). Part of the data, students studying healthcare programmes, has been taken from a study conducted by Tang et al. [15] to examine their compliance with infection control practices. In the present study, the majority of participants (87.9%) were from Putian. The distribution of age and gender and the details of the study programme by geographical location are presented in Table 1.

### 4.3. Ethical Considerations

Ethical clearance was obtained from the ethics committees of the participating institutions: Hong Kong (REC2020056) and Putian (2020-42). Participants were informed about the study and their rights before beginning the survey. Consent for the study was implied by returning the completed anonymous survey.

### 4.4. Measures

#### Perceived Stress

Feelings of stress were measured using a 10-item Perceived Stress Scale (PSS). According to Ng [16], the PSS represents two constructs, namely perceived helplessness and perceived self-efficacy. It includes 10 questions rated on a five-point scale, with a higher score indicating higher perceived stress level. The internal consistency of the PSS was strong, with a Cronbach’s alpha of 0.85 [17]. The PSS had an acceptable convergent validity with Pearson correlation coefficients of 0.69 and 0.72 [18].

### 4.5. Perceived Threat of COVID-19

Respondents’ perceptions of the health threat of the COVID-19 outbreak were assessed using a five-item Perceived Threat scale (PT). The scale was originally developed by Brug et al. [19] to assess the threat of the SARS epidemic. The PT is a four-point scale ranging from ‘not at all’ to ‘a great deal’; higher scores indicate a greater perceived threat. The internal reliability of the scale was acceptable (alpha = 0.65) [20].

### 4.6. Ways of Coping

Participants’ ways of coping were measured using items from the Ways of Coping Questionnaire [20]. Two subscales were used: Wishful Thinking Scale (WIS) and Empathic Responding Scale (EMP). In the present study, the WIS tested the extent to which respondents had managed their concerns or fears about COVID-19 through statements such as ‘wishing COVID-19 would go away or somehow be over with’ on a four-point scale ranging from ‘not at all’ to ‘a great deal’. The EMP tested the extent to which respondents had helped others who might be concerned about getting COVID-19, on a four-point scale ranging from ‘not at all’ to ‘a great deal’. The reliability of the scale was high (alpha = 0.91) [20].

### 4.7. COVID-19 Infection Control Practices

COVID-19 infection control practice (ICP) scores were measured using items developed by the SARS Collaborative Research Group [19,20]. The questionnaire consists of subscales measuring two aspects: (1) social distancing (SoD) scale (i.e., the practice of reducing close contact between people), and (2) personal precautionary measures (PPM) scale (i.e., the practice of maintaining good hygiene and lifestyle). The SoD scale measured behaviours in two domains: avoid going to public places and avoid contacting people who were perceived as having a higher risk of exposure to COVID-19. The first part—avoiding public places (APP)—comprises 10 possible behaviours, including statements such as ‘avoided travel to COVID-19 infected areas’ and ‘avoided eating in restaurants’. The second part—avoiding people (AP)—comprises nine items such as ‘avoided people who have a fever’ and ‘avoided a person you know who has just come from an area infected with COVID-19’. The reliability of the scale was moderately high, with alpha ranging from 0.79 to 0.91 [20,21].

The PPM scale asked respondents to identify health practices in which they had engaged to avoid being infected with COVID-19. There were eight health practices, such as ‘wearing a mask’ and ‘using disinfectants’. The scale’s reliability was moderately high (alpha = 0.78–0.83) [20,21,22]. The ICP score was used to represent the average of the APP, AP and PPM scores.

All questionnaires in this study were translated to Chinese for collecting data in Putian by forward-backward translation. The content validity (alpha = 0.7–1) of all scales was high. Internal consistency of the questionnaires in this study was established. The Cronbach alphas of perceived stress scale was 0.843; perceived threat scale was 0.718; ways of coping questionnaire was 0.849; COVID-19 infection control practices questionnaire was 0.927.

### 4.8. Data Analysis

All statistical analyses for this study were carried out using SPSS, version 26.0 (SPSS 2020). Descriptive statistics of frequencies, means and standard deviation were calculated for all variables. Chi-square χ^2^ and independent t-test were performed to test the statistical significance of differences in demographic characteristics and study variables between demographic groups (e.g., geographic locations). Pearson product-moment correlation was used to investigate the relationships between perceived stress, copings responses and ICP score. A three-step hierarchical multiple regression analysis was conducted. In each step, a group of predictors was entered to examine the contribution to the total variance of COVID-19 infection control practices. In the first step of analysis, we entered the demographic variables, including gender, age, institution, studying programme (e.g., nursing, engineering) and geographic location, etc. Dummy variables were created to incorporate categorical variables (e.g., gender, marital status) into the regression analyses. In the second step, we added perceived stress and threat. Finally, we entered coping variables (i.e., wishful thinking and empathetic responding) as predictors. For all statistical tests, variables were considered significant at a significance level of 0.05. Missing values, normality and outliers were checked before the main analysis; no outliers nor missing values were found in the dataset. Using G*Power [23], an a priori power analysis for linear multiple regression, fixed model, R^2^ increase with an alpha of 0.05, power of 0.08 and 14 predictors revealed that the sample size was sufficient to detect a small effect (f2 = 0.02).

## 5. Results

### 5.1. Mean Scores and Correlations of the Study Variables and Subgroup Analyses

The mean scores for PT (M = 11.79/20, SD = 2.99) and PSS (M = 19.55/40, SD = 4.69) indicated that the participants felt moderate stress towards, and threat from, COVID-19. The mean scores for WIS (M = 9.40/12, SD = 1.67) and EMP (M = 18.46/24, SD = 2.83) were high, which reflected the participants’ use of emotion-focused and relationship-focused coping in managing the stress. For COVID-19 infection control practices, the mean scores for APP and AP were 43.22 (SD = 8.05) and 37.43 (SD = 7.54), respectively; the mean score of PPM was 33.23 (SD = 5.18); and the mean score of ICP was 37.96 (SD = 5.55). There were significant differences between Hong Kong and Putian in all variables (*p* < 0.05). Participants in Hong Kong had a lower compliance in COVID-19 infection control practices. Table 1 displays the details of the descriptive variables.

Table 2 presents the study variables by programme and gender. Significant differences were observed in PT, AP and ICP scores between healthcare and non-healthcare programmes. Healthcare students had higher levels of threat and better compliance with social distancing by keeping away from people and performing personal precautionary measures. There were significant differences between gender in many of the variables (*p* < 0.05) but neither in perceived threat and empathetic response coping nor in personal precautionary measures. The male participants had a lower level of stress, less frequent use of wishful thinking and lower adherence to social distancing than that of the females. Further analyses found that men (*n* = 800) who studied in the junior year (*p* = 0.007) or studied in Putian (*p* = 0.011) had a higher level of stress; those who had higher education level (*p* = 0.05) or studied in Putian a (<0.001) had more frequent use of wishful thinking coping strategy; those who lived with family (*p* = 0.014) or in Putian (<0.001) tended to have better adherence to social distancing.

When gender differences were analysed by geographical location, significant differences were found in perceived stress and wishful thinking only. Females in Hong Kong had higher levels of perceived stress and often used wishful thinking. By contrast, in Putian, significant differences between genders were found in perceived threat and social distancing. Female participants exhibited higher levels of threat and showed better adherence to social distancing than males.

Table 3 shows the correlations between variables. Significant relationships were observed between perceived threat and stress. Significant negative relationships were found between perceived stress and avoiding public places and personal precautionary measures, indicating participants with lower levels of perceived stress were more likely to adhere to avoiding public places and personal precautionary measures. Coping responses were positively associated with overall infection control practices, meaning that participants who practiced more emotion-focused and relationship-focused coping tended to comply more with infection control practices. In analysing gender, significant negative relationships were detected between perceived stress and social distancing among males.

### 5.2. Multiple Hierarchical Regression Analysis for Variables Predicting COVID-19 Infection Control Practices Adherence

Table 4 presents the results of a three-step multiple hierarchical regression analysis of the associations between perceived threat and stress, coping responses and COVID-19-infection control practices, controlled for demographic conditions. All steps were significant (Step 1, F (10, 4380) = 8.483, *p* < 0.001); Step 2, F (12, 4378) = 8.251, *p* < 0.001); Step 3, F (14, 4376) = 30.407, *p* < 0.001). In Step 1, gender and geographical location predicted COVID-19-infection control practices and could explain 1.9% of variance in COVID-19 infection control practices, with which females and Putian participants displayed higher compliance. When perceived threat and stress were added to Step 2, the overall explanatory power increased to 2.2%, showing that perceived stress had predictive power for COVID-19 infection control practices. In Step 3, adding emotion-focused and relationship-focused coping responses significantly affected the model. There was an increase of 6.7% of adjusted R^2^, indicating that coping responses have potential to moderate COVID-19 infection control practices. Finally, the model explained 8.9% of the variance in COVID-19 infection control practices.

## 6. Discussion

In the present study, we found that university students demonstrated compliance with COVID-19 infection control practices. University students who are currently studying healthcare programmes tended to perceive more threat but complied better with the personal precautionary measures and social distancing to prevent viral infection, possibly owing to their knowledge of the COVID-19 infection and its detrimental effects on health. The model confirmed that gender, geographical location, perceived stress and emotion-focused and relationship-focused coping responses accurately predicted COVID-19 infection control practices. The most important finding from this study was that coping, whether emotion-focused or relationship-focused, had the greatest effect on infection control practices compared to other variables (i.e., perceived stress, gender and geographic location).

### 6.1. Geographic Location and Gender

This study showed that university students complied well with COVID-19 infection control practices. This finding is consistent with studies conducted in China reporting that university students showed high compliance with COVID-19 preventive measures [24,25]. Similar results were obtained from a European study on adolescents’ awareness of protective behaviour during the COVID-19 outbreak [26]. Residents of all ages in Hong Kong have been reported to practice a high level of self-protective behaviours during the pandemic [27]. It may be because this study assessed Chinese participants who were in the epicentre of the outbreak; they were probably more conscious about infection control measures to mitigate the spread of virus. This speculation is supported by Cassimatis et al. [28], who stated that Asian students were more aware of COVID-19 prevention guidelines established by the Centers for Disease Control and Prevention (CDC). Interestingly, in this study, university students in Putian reported better compliance with COVID-19 infection control practices than their counterparts in Hong Kong. It may be because there were more students studying healthcare programmes and more females in Putian than those in Hong Kong.

China was the first country to implement an aggressive lockdown strategy in Wuhan and other cities in January 2020. The infection control measures adopted by China included not only shutdowns of non-essential companies and shops and closures of public transport, airports and major highways, but also ‘closed management’ on a community basis in many areas across China to restrict social contacts. For instance, in some areas of China, only one family member was permitted to leave the household to purchase groceries. Access to villages and communities were prohibited throughout the day. Chinese authorities reported that there was no more domestic transmission of the disease after two months of the lockdown [29]. Conversely, in Hong Kong during that time, border crossings and public transport were still open. Regulations on restricting social contact, such as prohibition of group gatherings for more than four people, and temporary closures of entertainment venues were enacted only on 29 March 2020. The regulation of compulsory mask wearing in public places came into effect on 15 July 2020. Moreover, people’s positive perception of the government’s attempt to reduce the epidemic could increase practice of preventive measures [30,31]. In the present study, university students in Putian appeared to have stronger confidence in government policies controlling the outbreak.

Like other studies [25,28], females in this study complied with infection control practices more than males. This may be due to the gender norm that men are conditioned to be tough [32] and less likely to seek help from others or engage in health promotion activities [33]. This study also revealed that males who lived with family tended to have better adherence to social distancing. This may be due to their concern about spreading the virus to family members. Subgroup analysis showed that the gender differences with regard to infection control practices were mainly found in residents in Putian. Healthcare professionals should be aware of the masculine gender role that might undermine males’ willingness to engage in personal protective measures.

### 6.2. Perceived Threat and Stress

Our study confirmed that perceived stress is a predictor of infection control practices. The threat of COVID-19 was associated with perceived stress and had a negative relationship with personal precautionary measures. In this study, university students reported moderately high levels of perceived stress. This finding is consistent with previous studies that university students in China exhibited a certain degree of stress, experiencing stress-related symptoms, such as anxiety and depression [24,34], during the pandemic. In our study, university students in Hong Kong reported higher levels of stress than their counterparts in Putian. The finding is consistent with Dean et al.’s [35] study that people in Hong Kong exhibited the most psychological distress when compared to people in South Korea, the USA and France. Our study reports that female students in Hong Kong experienced higher stress levels. The many uncertainties surrounding COVID-19 generated stressors, such as the pressure for social distancing, lockdown, delays in academic achievement and economic uncertainty, all of which could heighten the stress level of an individual to varying degrees [1]. For university students in Hong Kong, the experience of social unrest in 2019 could also be one of the reasons behind higher stress levels [35,36].

The findings of the present study showed that perceived stress predicted infection control practices and was negatively associated with infection control practices. In contrast, other studies reported that higher levels of anxiety were more likely to initiate health behaviour when encountering health threats, such as novel swine-origin influenza [8,10]. In our study, we assessed the perceived stress level with the PSS scale, which comprises perceived helplessness and perceived self-efficacy. Self-efficacy is defined as the self-confidence necessary to engage in behaviour to successfully achieve a task [37]. A lower level of self-efficacy (i.e., high level of perceived stress) reflects lack of confidence in managing the situation. Our study suggests that confidence predicts behavioural changes.

Female university students with lower levels of stress are more likely to perform personal precautionary measures, such as wearing masks, practicing hand hygiene, social distancing and avoiding public places. In contrast, male university students with lower levels of stress are more likely to wear masks and wash their hands only. This may be due to the fact that females have less preferences for outdoor activities than males [38]. These findings indicate that to mitigate the spread of any virus, or other community threat to health, a psychology-oriented and gender-sensitive programme should be implemented.

### 6.3. Emotion-Focused and Relationship-Focused Coping

In our study, the results showed that university students often used emotion-focused and relationship-focused coping to deal with stressful situations during COVID-19. This finding is consistent with previous studies that found that emotional coping was frequently used to manage stressful situations during the COVID-19 outbreak [14,39]. The present study showed an association between coping responses and stress. The respondents employed emotion- and relationship-focused forms of coping to manage their emotions in the low-control situation of COVID-19. Kulenović and Buško [40] similarly found that high levels of perceived stress predict the frequent use of avoidance as a coping strategy. Emotion-focused coping is considered to be maladaptive [12]. However, the emotion- and relationship-focused coping strategies used by the university students in this study seemed to be adaptive. The respondents were motivated to protect themselves in an uncontrollable public health situation. This adaptive function was also noted during the SARS epidemic [41].

In this study, both wishful thinking and empathetic responding coping strategies may have a moderating effect on COVID-19 infection control practices. These findings were inconsistent with previous studies on epidemics [20,21]. The many unknowns of treatment protocol and protective effects of vaccines for the new variants of the current virus [42] may explain the difference in findings between the previous studies and current research. People may turn to wishful thinking and empathetic responding to relieve their psychological distress when facing uncertainty. Further, during periods of social distancing, people are likely to experience feelings of isolation. Lonely people tend to develop empathy as an adaptive emotion-regulating strategy to reduce their loneliness [43]. Use of these coping strategies may have protective effects on mental health [25]. Our study confirms the use of these coping strategies appeared to have a direct effect on infection control practices during the pandemic. This may be explained by the mediating effect of coping on behavioural responses [40]. The current study suggests that coping appraisal is a necessary factor motivating people to initiate protective actions [44]. These findings support the implementation of interventions that increase the effectiveness of coping strategies and thereby promote positive health behaviours. Thus, public health interventions aimed at university students should take emotion- and relationship-focused coping strategies into account. When sending outbreak information to university students, strategies can be designed to motivate their empathetic feelings and increase their sense of control of, or power in, the situation.

### 6.4. Strengths and Limitations

The strength of this study is the large number of sample sizes that provided more accurate mean values. Another strength is that the study was conducted in the early stage of the pandemic (April 2020), which provided a more accurate picture on the psychological and behavioural responses towards a disease outbreak with unknown causes. This study had several important limitations. First, the variance explained by the independent variables was relatively small. Future studies are recommended to identify factors that could explain larger proportions of the variance in COVID-19 infection control practices. Second, the results of the regression analysis were based on correlational data, meaning that it is not possible to draw firm conclusions about causal relationships between these variables. Third, this study relied on self-reported measures, so the results may have been affected by social desirability bias. Fourth, the unequal sample sizes in gender may undermine the between group comparison. Fifth, as all the participants enrolled in this study were Chinese undergraduates, the results may not be generalisable to other countries with different cultures and socio-economic characteristics. In view of the global nature of COVID-19, future studies should include a broader sample.

## 7. Conclusions

This study found that gender, geographic location, perceived stress, wishful thinking and empathetic responding are predictors of COVID-19 infection control practices. These findings should be considered in the current situation or in models of potential future pandemics. This study found that male university students tended to comply less with social distancing. The development of public health measures should recognise the importance of gender differences. Relevant government departments should tailor public health measures to local needs. Our study revealed that university students displayed moderate levels of stress in the face of the pandemic. Healthcare providers and educators should be aware that wishful thinking and empathetic responding are coping strategies this demographic uses to preserve their psychological well-being during a pandemic. Therefore, counselling services should emphasise reassurance and empathy; additional emotional support should be provided to alleviate anxieties. This study also revealed factors related to noncompliance with COVID-19 infection control practices. Given the global effects of COVID-19 on health, social functioning and economic stability, strategies addressing noncompliance with public health measures are critical to mitigate further transmission.

## Figures and Tables

**Table 1 ijerph-19-05327-t001:** Demographic characteristics of the participants and study variables by geographical location.

		Hong Kong	Fujian	Total	
		(*n* = 531)	(*n* = 3861)	(*n* = 4392)	
		*n* (%)/M (SD)			*p*-Value
Age		21.09 (2.71)	20.45 (1.47)	20.5 (1.68)	<0.001 ^a^ ***
Gender					<0.001 ^b^ ***
	Female	390 (73.4)	3202 (82.9)	3592 (81.8)	
	Male	141 (26.6)	659 (17.1)	800 (18.2)	
Academic level of the study programme					<0.001 ^b^ ***
	Diploma	3 (0.6)	0 (0.0)	3 (0.1)	
	Higher Diploma	93 (17.5)	0 (0.0)	93 (2.1)	
	Associate Degree	3 (0.6)	1380 (35.7)	1383 (31.5)	
	Bachelor Degree	432 (81.4)	2481 (64.3)	2913 (66.3)	
Study programme					<0.001 ^b^ ***
	Nursing	397 (74.8)	2099 (54.4)	2496 (56.8)	
	Engineering	0 (0.0)	391 (10.1)	391 (8.9)	
	Commerce	0 (0.0)	293 (7.6)	293 (6.7)	
	Medical Science	23 (4.3)	242 (6.3)	265 (6.0)	
	Management studies	2 (0.4)	204 (5.3)	206 (4.7)	
	Arts	0 (0.0)	142 (3.7)	142 (3.2)	
	Science	0 (0.0)	137 (3.5)	137 (3.1)	
	Early Childhood Education	9 (1.7)	113 (2.9)	122 (2.8)	
	Medical Laboratory Science	22 (4.1)	48 (1.2)	70 (1.6)	
	Liberal Arts	0 (0.0)	65 (1.7)	65 (1.5)	
	Chinese Medicine	0 (0.0)	48 (1.2)	48 (1.1)	
	Radiation Therapy	11 (2.1)	21 (0.5)	32 (0.7)	
	Pharmacy	0 (0.0)	28 (0.7)	28 (0.6)	
	Health Science	21 (4.0)	0 (0.0)	21 (0.5)	
	Occupational Therapy	18 (3.4)	0 (0.0)	18 (0.4)	
	Physiotherapy	14 (2.6)	0 (0.0)	14 (0.3)	
	Gerontology	7 (1.3)	0 (0.0)	7 (0.2)	
	Psychology	7 (1.3)	0 (0.0)	7 (0.2)	
	Others	0 (0.0)	30 (0.8)	30 (0.7)	
Year of study					<0.001 ^b^ ***
	Year 1	162 (30.5)	1906 (49.4)	2068 (47.0)	
	Year 2	122 (23.0)	1038 (26.9)	1160 (26.0)	
	Year 3	134 (24.7)	737 (19.1)	868 (20.0)	
	Year 4	72 (13.6)	159 (4.1)	231 (5.0)	
	Year 5	43 (8.1)	21 (0.5)	64 (1.0)	
Clinical experience					<0.001 ^b^ ***
	No clinical experience	139 (26.2)	266 (6.9)	3328 (75.8)	
	Less than 12 weeks	158 (29.8)	501 (13.0)	405 (9.2)	
	More than 12 weeks	234 (44.1)	3094 (80.1)	659 (15.0)	
Study variables					
	Perceived threat (PT)	15.65 (3.12)	11.26 (2.55)	11.79 (2.99)	<0.001 ^a^ ***
	Perceived stress (PSS)	21.42 (5.09)	19.30 (4.58)	19.55 (4.69)	<0.001 ^a^ ***
Coping response					
	Wishful thinking (WIS)	9.57 (2.11)	9.37 (1.59)	9.40 (1.67)	0.036 ^a^ *
	Empathetic responding (EMP)	17.88 (3.85)	18.54 (2.65)	18.46 (2.83)	<0.001 ^a^ ***
Social distancing (SoD)					
	Avoiding public places (APP)	41.57 (6.28)	43.44 (8.24)	43.22 (8.05)	<0.001 ^a^ ***
	Avoiding people (AP)	35.99 (7.95)	37.62 (7.46)	37.43 (7.54)	<0.001 ^a^ ***
Personal precautionary measures (PPM)		31.89 (4.72)	33.41 (5.21)	33.23 (5.18)	<0.001 ^a^ ***
COVID-19 infection control practices (ICP)		36.49 (4.81)	39.16 (5.61)	37.96 (5.55)	<0.001 ^a^ ***

^a^ Independent *t*-test; ^b^ Chi-square test; * *p* < 0.05.; *** *p* < 0.001; M = mean; SD = standard deviation.

**Table 2 ijerph-19-05327-t002:** Study variables by programme, gender, and by gender per geographical location.

		Healthcare Programme		Non-Healthcare Programme		Total		
		(*n* = 2706)		(*n* = 1686)		(*n* = 4392)		
		M (SD)						*p*-Value ^a^
Perceived threat (PT)		11.98 (3.13)		11.49 (2.73)		11.79 (2.99)		<0.001
Perceived stress (PSS)		19.58 (4.64)		19.51 (4.78)		19.55 (4.69)		0.604
Coping response								
	Wishful thinking (WIS)	9.40 (1.68)		9.38 (1.64)		9.40 (1.67)		0.718
	Empathetic responding (EMP)	18.45 (2.85)		18.48 (2.79)		18.46 (2.83)		0.708
Social distancing (SoD)								
	Avoiding public places (APP)	43.28 (8.18)		43.11(7.84)		43.22 (8.05)		0.488
	Avoiding people (AP)	37.76 (7.48)		36.89 (7.60)		37.43 (7.54)		<0.001
Personal precautionary measures (PPM)		33.41 (5.04)		32.93 (5.38)		33.23 (5.18)		0.003
COVID-19 infection control practices (ICP)		38.15 (5.52)		37.64 (5.58)		37.96 (5.55)		0.003
		**Male**		**Female**		**Total**		
		**(*n* = 800)**		**(*n* = 3592)**		**(*n* = 4392)**		
		**M (SD)**						***p*-Value ^a^**
Perceived threat (PT)		11.81 (3.43)		11.79 (2.89)		11.79 (2.99)		0.873
Perceived stress (PSS)		19.20 (5.01)		19.63 (4.62)		19.55 (4.69)		0.024
Coping response								
	Wishful thinking (WIS)	9.26 (1.89)		9.42 (1.61)		9.40 (1.67)		0.026
	Empathetic responding (EMP)	18.48 (3.35)		18.46 (2.70)		18.46 (2.83)		0.886
Social distancing (SoD)								
	Avoiding public places (APP)	41.86 (8.70)		43.52 (7.87)		43.22 (8.05)		<0.001
	Avoiding people (AP)	36.00 (8.36)		37.74 (7.30)		37.43 (7.54)		<0.001
Personal precautionary measures (PPM)		32.99 (6.00)		33.28 (4.97)		33.23 (5.18)		0.207
COVID-19 infection control practices (ICP)		37.99 (6.59)		39.28 (5.57)		37.96 (5.55)		<0.001
		**Hong Kong**			**Fujian**			
		**(*n* = 531)**			**(*n* = 3861)**			
		**Male**	**Female**		**Male**	**Female**		**Total**
		**(*n* = 141)**	**(*n* = 390)**		**(*n* = 659)**	**(*n* = 3202**)		**(*n* = 4392)**
		**M (SD)**		***p*-Value ^a^**	**M (SD)**		***p*-Value ^a^**	
Perceived threat (PT)		15.57 (3.30)	15.67 (3.06)	0.745	11.00 (2.87)	11.31 (2.48)	0.010	11.79 (2.99)
Perceived stress (PSS)		20.16 (5.55)	21.88 (4.83)	0.001	18.99 (4.86)	19.36 (4.51)	0.074	19.55 (4.69)
Coping response								
	Wishful thinking (WIS)	8.88 (2.35)	9.82 (1.96)	<0.001	9.35 (1.76)	9.38 (1.56)	0.682	9.40 (1.66)
	Empathetic responding (EMP)	17.62 (3.94)	17.97 (3.81)	0.362	18.66 (3.18)	18.50 (2.73)	0.285	18.46 (2.83)
Social distancing (SoD)								
	Avoiding public places (APP)	41.06 (7.35)	41.76 (5.85)	0.305	42.04 (8.96)	43.73 (8.06)	<0.001	43.23 (8.04)
	Avoiding people (AP)	35.46 (8.65)	36.19 (7.69)	0.353	36.12 (8.30)	37.93 (7.23)	<0.001	37.43 (7.53)
Personal precautionary measures (PPM)		31.84 (5.53)	31.91 (4.39)	0.883	33.24 (6.07)	37.93 (7.24)	0.413	33.23 (5.18)
COVID-19 infection control practices (ICP)		36.12 (5.70)	36.62 (4.44)	0.345	37.13 (6.44)	38.37 (5.40)	<0.001	37.96 (5.54)

^a^ Independent *t*-test; M = mean; SD = standard deviation.

**Table 3 ijerph-19-05327-t003:** Correlation matrix between perceived threat, perceived stress, coping responses and COVID-19 infection control practices in all participants.

	Total Sample							
		2	3	4	5	6	7	8
1. Perceived threat (PT)		0.273 ***	0.086 ***	0.000	−0.025	−0.001	−0.067 ***	−0.033 *
2. Perceived stress (PSS)			0.133 ***	−0.034 *	−0.035 *	−0.014	−0.122 ***	−0.061 ***
3. Wishful thinking (WIS)				0.353 ***	0.111 ***	0.163 ***	0.146 ***	0.173 ***
4. Empathetic responding (EMP)					0.170 ***	0.168 ***	0.281 ***	0.246 ***
5. Avoiding public places (APP)						0.553 ***	0.348 ***	0.842 ***
6. Avoiding people (AP)							0.416 ***	0.850 ***
7. Personal precautionary measures (PPM)								0.668 ***
8. COVID-19 infection control practices (ICP)								
	**Correlation Matrix by Gender**							
	1	2	3	4	5	6	7	8
1. Perceived threat (PT)	-	0.273 ***	0.097 ***	0.008	−0.034 *	−0.009	−0.075 ***	−0.044 **
2. Perceived stress (PSS)	0.277 ***	-	0.139 ***	−0.030	−0.033 *	−0.014	−0.107 ***	−0.062 ***
3. Wishful thinking (WIS)	0.050	0.105 **	-	0.357 ***	0.085 ***	0.151 ***	0.144 ***	0.156 ***
4. Empathetic responding (EMP)	−0.027	−0.044	0.342 ***	-	0.149 ***	0.141 ***	0.269 ***	0.221 ***
5. Avoiding public places (APP)	0.006	−0.057	0.188 ***	0.243 ***	-	536 ***	0.318 ***	0.835 ***
6. Avoiding people (AP)	0.024	−0.029	0.193 ***	0.259 ***	0.598 ***	-	0.406 ***	0.846 ***
7. Personal precautionary measures (PPM)	−0.041	−0.109 **	0.148 ***	0.319 ***	0.445 ***	0.448 ***	-	0.653 ***
8. COVID-19 infection control practices (ICP)	0.000	−0.073 *	0.218 ***	0.326 ***	0.863 ***	0.856 ***	0.718 ***	-

Correlations for male are below the diagonal, and correlations for female are above the diagonal. * *p* < 0.05; ** *p* < 0.01; *** *p* < 0.001.

**Table 4 ijerph-19-05327-t004:** Predictors of COVID-19 infection control practices adherence using multiple hierarchical regression.

	Step 1		Step 2		Step 3	
	B	β	B	β	B	β
Age	−0.036	−0.011	−0.041	−0.013	−0.006	−0.002
Gender	−0.971	−0.068 ***	−0.996	−0.069 ***	−0.993	−0.069 ***
Institution	0.329	0.039	0.357	0.042	0.289	0.034
Level of study programme	−0.133	−0.016	−0.116	−0.014	−0.142	−0.017
Studying programme	0.018	0.011	0.016	0.010	0.024	0.015
Year of study	0.083	0.015	0.093	0.017	0.070	0.013
Clinical experience	−0.180	−0.021	−0.164	−0.019	−0.137	−0.016
Marital status	−2.409	−0.019	−2.259	−0.017	−1.379	−0.011
Living condition	0.843	0.019	0.773	0.018	0.836	0.019
Geographical location	−1.101	−0.065 **	−1.157	−0.068 **	−0.872	−0.051 *
Perceived threat (PT)			0.051	0.033	0.019	0.010
Perceived stress (PSS)			−0.067	−0.057 ***	−0.074	−0.063 ***
Wishful thinking (WIS)					0.364	0.109 ***
Empathetic responding (EMP)					0.391	0.199 ***
**R^2^**	0.019		0.022		0.089	
**Adjusted R^2^**	0.017		0.019		0.086	
**R^2^ change**	0.019		0.003		0.067	
**F**	8.483 ***		8.251 ***		30.407 ***	

Step 1: df1 = 10, df2 = 4380; Step 2: df1 = 12, df2 = 4378; Step 3: df1 = 14, df2 = 4376; B = standardized regression estimates; β = unstandardized regression estimate; Gender: value 1 = male; 0 = female; Marital Status: value 1 = married; 0 = unmarried; Geographical location: value 1 = Hong Kong; Fujian = 0; * *p* < 0.05; ** *p* < 0.01; *** *p* < 0.001.

## Data Availability

Not applicable.

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
