# Peer review of "Compliance with Standard Precaution and Its Relationship with Views on Infection Control and Prevention Policy among Chinese University Students during the COVID-19 Pandemic"

_ijerph, 2022, doi:10.3390/ijerph19095327_

Round 1
Reviewer 1 Report
Dear authors,
This is very interesting research, I read your manuscript very carefully. It is very well organized, and data are presented in very clear way.
I have some questions:
Why did you decide to use only two subscales in Ways of Coping Questionnaire? It would be interesting to see results of other subscales.
Please provide internal consistency and reliability of used questionnaires on your sample.
Women are majority group in your sample (81.8%) - please explain how justified was to observe differences between men and women and to do other statistical procedures on men in observed variables and to make some conclusions?
Author Response
|
Many thanks for the constructive comments provided by the reviewer. Please find the responses to the comments below.
|
|
|
Comment 1 |
Why did you decide to use only two subscales in Ways of Coping Questionnaire? It would be interesting to see the results of other subscales. |
|
Response |
We agreed that it would be interesting to see the results of other subscales, but we tended to collect data needed to address the research questions. In view of the aim of this study, we sought to examine ways in which the perception of the threat of COVID-19 was related to coping, and in turn, how coping was related to health behaviors, we chose the WIS and EMP to represent the ways of coping in this pandemic. We expected specific coping strategies are involved in facilitating specific health behaviors. Given the avoidant nature of wishful thinking as a coping strategy, it seems reasonable to assume that the use of such a strategy for managing the threat of COVID-19 might be associated with avoidance-type health behaviors, such as social distancing. As empathic responding involves considering this stressful experience of COVID-19 in terms of other’s well-being. Given the use of empathy, one might consider other people needing and requiring care. It is assumed that efforts for engaging in health behaviors involve more preventative and effective health behaviors such as hand washing. |
|
Comment 2 |
Please provide internal consistency and reliability of used questionnaires on your sample. |
|
Response |
The reliability of the questionnaires is estimated by internal consistency. The Cronbach alphas of PSS =0.843; PT=0.718; way of coping=0.849; COVID-19 Infection Control Practices =0.927. The information has been added to the end of the methods section (line 149-151). |
|
Comment 3 |
Women are majority group in your sample (81.8%) - please explain how justified was to observe differences between men and women and to do other statistical procedures on men in observed variables and to make some conclusions? |
|
Response |
Thanks and we agreed that the unequal sample size may undermine the results. Though we used the independent samples t-tests can handle unequal sample sizes when used to compare the differences between mean, the observed differences between men and women may be limited by the unequal sample size. This would be addressed as a limitation (line 360). Further statistical analyses were conducted on men to enrich the results, i.e., further analyses found that men (n=800) who studied in the junior year (p=0.007) or studied in Putian (p=0.011) had a higher level of stress; those who had higher education level (p=0.05) or studied in Putian a (<0.001) had more frequent use of wishful thinking coping strategy; who lived with family (p=0.014) or in Putian (<0.001) tended to have a better adherence to social distancing. The discussion was enriched with information that “this study also revealed that males who lived with family tended to have a better adherence to social distancing. This may be due to concern of spreading the virus to family members”. The information was included in the result section (line 198-202) and the discussion section (line 296).
|
Reviewer 2 Report
- Line 63, the objectives "were"? why not are
- section2, line 140. on what basis was the multiple regression analysis was chosen? how would this method specifically serve the purpose of the paper/or objective.
- the study is built on variables from theories such as protection motivation theory and transactional theory along with variables from the body of literature. The authors immediately discuss the method instead of explaining where do these different variables come from and how they were put together. This can be fixed by including a literature section of background section.
- section 5, it is worth mentioning the limitation of this study that could be related to the generalization of results or in gender % among respondents.
Author Response
|
Many thanks for the constructive comments provided by the reviewer. Please find the responses to the comments below. |
|
|
Comment 1 |
Line 63, the objectives "were"? why not are
|
|
Response |
The manuscript reported events in the past, therefore, past tense was used. The manuscript was edited by a native English editor who confirmed the use of language was correct. |
|
Comment 2 |
section2, line 140. on what basis was the multiple regression analysis was chosen? how would this method specifically serve the purpose of the paper/or objective. |
|
Response |
A multiple regression analysis is a type of test that analyzes the amount of variance explained in a dependent variable by more than one predictor variable. This study involved multiple predictors including demographic characteristics, perceived stress, and copings etc. to explain the variance of the dependent variable, i.e., COVID-19 infection control practices adherence. This statistical method can therefore achieve the objective (b) identifying the factors that could be able to predict the COVID-19 infection control practices adherence of this study. [Reference: Jeong, Y., & Jung, M. J. (2016). Application and interpretation of hierarchical multiple regression. Orthopaedic Nursing, 35(5), 338-341.] |
|
Comment 3 |
the study is built on variables from theories such as protection motivation theory and transactional theory along with variables from the body of literature. The authors immediately discuss the method instead of explaining where do these different variables come from and how they were put together. This can be fixed by including a literature section of background section.
|
|
Response |
A section on review of the theoretical constructs, i.e., threat and behavioral responses, stress, ways of coping was added. (line 46-73). |
|
Comment 4 |
section 5, it is worth mentioning the limitation of this study that could be related to the generalization of results or in gender % among respondents.
|
|
Response |
Thank you and agree. Limitation on generalization was reported (line 382). Limitation about unequal sample sizes was added (line 380) |
Reviewer 3 Report
This work examines the relationships between the perceived threat, perceived stress, coping responses and infection control practices regarding the COVID-19 pandemic among university students in China
The manuscript is well organized, an adequate approach has been carried out, and the data has been correctly presented and discussed by the Authors.
Small corrections:
Abstract:
Please provide a small introduction.
Please detail, in terms of a socio-economical context, why Hong Kong and Putian populations have different behaviours in infection control practice.
Conclusion
Line 355
“This section is mandatory. This study found that gender, geographic location, perceived stress, wishful thinking and empathetic responding are predictors of COVID-19 infection control practices.”
Please delete “this section is mandatory”
Author Response
|
Many thanks for the constructive comments provided by the reviewer. Please find the responses to the comments in below table. The manuscript has been revised and highlighted in yellow. |
|
|
Comment 1 |
Abstract: Please provide a small introduction. |
|
Response |
An introduction was added as background : COVID-19 has placed tremendous pressure on the global public health system and has changed daily life. |
|
Comment 2 |
Please detail, in terms of a socio-economical context, why Hong Kong and Putian populations have different behaviours in infection control practice. |
|
Response |
The behaviours in infection control practice, in terms of the socio-economical context, that it may be because there were more students studying healthcare programs and more females in Putian than those in Hong Kong were further explained (line 274). |
|
Comment 3 |
Conclusion Line 355 “This section is mandatory. This study found that gender, geographic location, perceived stress, wishful thinking and empathetic responding are predictors of COVID-19 infection control practices.” Please delete “this section is mandatory”
|
|
Response |
Thanks for reminding me. The phrase “this section is mandatory” was deleted. |
Round 2
Reviewer 1 Report
Dear authors,
thank you for your explanations and for taking into consideration my review.
I don't have any suggestions.